behaviour/ecology

flight speed, predation risk, protean behaviour, flocking, landscape, pigeon

# Fine-scale changes in speed and altitude suggest protean movements in homing pigeon flights

Baptiste Garde[1], Rory P. Wilson[1], Emmanouil Lempidakis[1], Luca Börger[1], Steven J. Portugal[2], Anders Hedenström[3], Giacomo Dell'Omo[4], Michael Quetting[5], Martin Wikelski[5,6] and Emily L. C. Shepard[1]

[1]Biosciences, College of Science, Swansea University, Singleton Park, Swansea, UK
[2]Department of Biological Sciences, Royal Holloway University of London, Egham, UK
[3]Department of Biology, Centre for Animal Movement Research, Lund University, Lund, Sweden
[4]Ornis italica, Rome, Italy
[5]Centre for the Advanced Study of Collective Behaviour, University of Konstanz, 78457 Konstanz, Germany
[6]Department of Migration and Immuno-Ecology, Max Planck Institute of Animal Behavior, Radolfzell, Germany

BG, 0000-0002-8726-6279; RPW, 0000-0003-3177-0107;
EL, 0000-0003-2384-9093; LB, 0000-0001-8763-5997;
SJP, 0000-0002-2438-2352; AH, 0000-0002-1757-0945;
GD, 0000-0002-9601-9675; MW, 0000-0002-9790-7025;
ELCS, 0000-0001-7325-6398

The power curve provides a basis for predicting adjustments that animals make in flight speed, for example in relation to wind, distance, habitat foraging quality and objective. However, relatively few studies have examined how animals respond to the landscape below them, which could affect speed and power allocation through modifications in climb rate and perceived predation risk. We equipped homing pigeons (*Columba livia*) with high-frequency loggers to examine how flight speed, and hence effort, varies in relation to topography and land cover. Pigeons showed mixed evidence for an energy-saving strategy, as they minimized climb rates by starting their ascent ahead of hills, but selected rapid speeds in their ascents. Birds did not modify their speed substantially in relation to land cover, but used higher speeds during descending flight, highlighting the importance of considering the rate of change in altitude before estimating

**Author for correspondence:**
Baptiste Garde
e-mail: baptiste.garde@swansea.ac.uk

power use from speed. Finally, we document an unexpected variability in speed and altitude over fine scales; a source of substantial energetic inefficiency. We suggest this may be a form of protean behaviour adopted to reduce predation risk when flocking is not an option, and that such a strategy could be widespread.

## 1. Introduction

Time and energy are currencies that have a profound influence on animal movement, with the judicious use of energy being particularly pertinent for flying animals, due to the scale of the costs in flapping flight [1,2]. Indeed, in-flight decisions such as route choice [3,4], flight altitude [4] or speed [5–7] can markedly affect power consumption on a second-by-second basis.

Flight speed is particularly relevant with regard to energy expenditure because the power required for flight is predicted to follow a U-shaped curve, from a high point during hovering, down through a minimum, to an exponentially increasing power load with increasing speed thereafter [5]. This power curve can be used to predict a range of optimal speeds including the minimum power speed ($V_{mp}$), which requires the least energy per unit time, and the maximum range speed ($V_{mr}$), which uses the least energy per unit distance travelled [5]. Observations indicate that most birds travel at speeds between $V_{mp}$ and $V_{mr}$ [8,9], with the specific predictions often borne out according to the situation and the purpose of the flight (e.g. display flight versus foraging or migration, etc. [7,10,11]). Optimal flight speeds are also predicted to vary with head- and tailwinds [11], and a range of studies show that birds adjust their airspeeds accordingly [12–14]. Finally, birds should reduce their airspeed as they climb, in line with the increase in energy required to gain potential energy [13,15].

There are, however, instances where birds fly at speeds above $V_{mr}$. Faster travel can be achieved for a minimal cost when birds fly at their minimum time speed ($V_{mt}$) [11,13]. Circumstances may also favour non-energy-efficient speeds, for instance, faster flight during foraging may increase the provisioning rate of hatchlings [16–18] and speed can be advantageous during migration if birds then arrive at the breeding grounds before competitors, increasing the likelihood of reproductive success [19]. Birds can also vary their speed when flying in a group [20] compared with when they fly solo, if the benefits of maintaining group cohesion outweigh the costs of flying at speeds that are suboptimal for energy use [14,21].

Overall, flight speed seems, therefore, to vary with (i) the currency that is driving the movement, and (ii) the physical environment, which impacts the efficiency of any given speed. However, studies examining both of these factors tend to quantify speed at relatively large scales, averaging it over individual flights or large sections of the track (e.g. [13,22], though see [9,23]). This means that factors impacting the choice of flight speed over fine scales, including changes in the substrate (mainly land cover and topography) that birds fly over, tend to be averaged out. Land cover could first affect birds directly, due to the way that the substrate affects the movement of air above it, with some land types more likely to generate rising air, for instance [24]. The land cover might also affect flight indirectly, as different habitats present different predation risks. For instance, pigeons are more likely to be attacked by peregrine falcons (*Falco peregrinus*) swooping from above in open spaces, while woodlands can be associated with goshawks (*Accipiter gentilis*) attacking from below [25]), or waiting for them next to their loft [26].

We released solo-flying homing pigeons (*Columba livia*), equipped with high-frequency GPS and pressure sensors, to examine the extent to which a flapping flier modulates its airspeed within individual flights, and specifically in relation to the substrate. Pigeons have been the dominant model species used in studies examining navigation mechanisms, which are strongly linked to landscape features over fine scales [27–30]. Nonetheless, there have been no studies assessing whether the landscape affects their speed, or the resulting implications for energy efficiency and predation avoidance. Homing pigeons have been selected for racing and are thus expected to invest primarily into speed during their homing flights. However, we expected birds to reduce speed when climbing [13]. We, therefore, predicted that the greatest changes in speed would depend on the topography, with individuals decreasing their airspeed with increased climb rate (cf. [13,31]). We also assessed whether birds minimize their climb rate by climbing gradually ahead of a high point, or whether they track the terrain beneath them (resulting in higher instantaneous climb rates). We also quantified variation in speed in relation to land cover, predicting that an increase in speed or altitude above a certain type of land cover is likely to represent a response to greater perceived predation risk. Overall, this should provide insight into the fine-scale changes in effort and perceived risk driven by the landscape that could ultimately influence the costs associated with route choice when a flight is considered in all three dimensions.

# 2. Material and methods

Homing pigeons (rock doves, *C. livia* Linnaeus) were equipped with high-frequency GPS loggers linked to barometric pressure sensors (see below) and released on solo homing flights from Bodman-Ludwigshafen in Germany (47.815° N, 8.999° E, figure 1), between 24 and 31 July 2018 and 3 and 19 April 2019. The release site was an open field 5.7 km north of their home loft. Releases were conducted during the morning, in weather ranging from sunny to cloudy and in a range of temperatures from 23° to 36°C in July, and from 7° to 19°C in April. Every day, six pigeons were brought to the release site by car, in a common transport box preventing them from seeing outside. Birds were taken out of the transport box 2 min before the release. Changes in homing efficiency in response to route familiarity can still be observed 20 flights after the first release [32]. Pigeons were, therefore, flown with dummy loggers from the release site more than 30 times prior to trials [21,32,33] to remove changes associated with route learning (a phase also associated with increased inter-individual variability linked to differences in learning and navigational capabilities, as well as personality [34,35]). The same birds were used in 2018 and 2019, with bird masses, wingspan and wing area taken once for each release session. Wing loading was calculated as the ratio of body mass to wing area, following Pennycuick [8]. R package 'afpt' [36] was used to calculate the theoretical minimum power speed ($V_{mr}$) and maximum range speed ($V_{mp}$) based on those measurements and a body drag coefficient of 0.2 [36].

Birds were equipped with two data loggers: a Daily Diary (Wildbyte Technologies, Swansea University, UK) and a GPS (GiPSy 5, Technosmart Europe, Guidonia-Montecelio, Italy). The Daily Diaries recorded a range of parameters including pressure at 4 Hz (using Bosch pressure sensor BMP280 with a relative accuracy of ±0.12 hPa, equivalent to ±1 m), while the GPS was set to sample at 1 Hz for the July flights and 5 Hz for April flights (data were subsequently subsampled to 1 Hz). The two units were connected to each other and the Daily Diary was programmed to receive an initial timestamp from the GPS in order to synchronize the time between the two datasets. Loggers were combined in a lightweight, 3D printed housing, producing a unit measuring $47 \times 22 \times 15$ mm and weighing up to 18.0 g [14,32], and representing between 3.8% and 4.2% of a bird's body mass. Loggers were attached to the back of the bird via Velcro strips, with the bottom strip being glued to the pigeon's back feathers [37]. All procedures were approved by the Swansea University Animal Welfare and Ethical Review Body (AWERB) (approval number: IP-1718-23) and by the Regierungspräsidium Freiburg (reference number: G-17/92).

An anemometer (Kestrel 5500 L, Kestrel instruments, USA) was deployed in an open location at the release site on a 5 m pole and set to record wind speed and direction every 10 s. Flights with an average wind speed greater than 2 m s$^{-1}$ were not used in the analysis in order to control for the influence of wind on the selection of flight speed, which is already well established (e.g. [11,38]). In addition, circling was identified in the GPS tracks and excluded from the analysis [39,40]. Resting was also excluded from the flight, along with the descent before landing and the ascent after take-off.

The 2012 Corine Land Cover classification (100 m resolution, land.copernicus.eu) was used to determine two categories of land cover; open land (which mainly constituted fields in our study area, figure 1) and woodland. Elevation data were obtained from a digital surface model (DSM) (30 m resolution, source: https://opendem.info/index.html). The topography of the area between the release site and the loft included a valley, and flights were classified according to whether they were routed along the valley (where ground elevations were less than 465 m) or over the hill (where flight altitudes exceeded 465 m, figure 1).

Flight altitude above mean sea level (ASL) was calculated by smoothing the barometric pressure data over 5 s to reduce any potential noise caused by the wingbeats and the pressure sensor, and converting pressure to altitude adjusting for daily changes in pressure at the release site in the seconds preceding take-off. The barometric pressure was used to estimate altitude, due to greater within-flight accuracy [41]. Altitude above ground level was calculated as the difference between flight altitude and the elevation of the substrate. Groundspeed and heading were calculated from consecutive GPS fixes, using the R package 'move' v. 3.1.0 [42], and smoothed over 5 s to reduce GPS error. The speed of the bird relative to the horizontal movement of the surrounding air, or horizontal airspeed $V_x$ (m s$^{-1}$), was taken as

$$V_x = \sqrt{V_g^2 + V_w^2 + 2 V_g V_w \cos\left(\frac{\theta \times \pi}{180}\right)}, \tag{2.1}$$

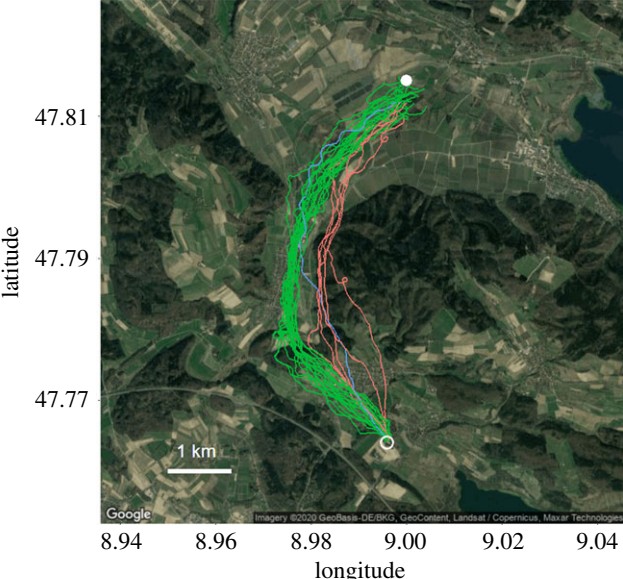

**Figure 1.** GPS tracks of 29 pigeon homing flights (seven individuals) from the release site (filled circle) to the loft (empty circle). Green tracks correspond to 'valley' flights ($n = 20$), red tracks to 'hill' flights ($n = 5$) and the blue track corresponds to a flight that started over the valley but reached the hill towards the end.

where $V_g$ is the groundspeed, $V_w$ is the wind speed and $\theta$ is the angle between the bird heading and the wind direction (ranging between 0° and 180°). These values were then adjusted to account for the climb rate, giving airspeed, $V_a$, as the vector sum of the horizontal airspeed $V_x$ and the climb rate $V_z$

$$V_a = \sqrt{V_x^2 + V_z^2}. \tag{2.2}$$

Finally, the rate of change of speed (i.e. forward acceleration and deceleration) and altitude (i.e. climb rate, $V_z$) were calculated as the difference between consecutive estimates of speed (at 1 Hz) or altitude (smoothed values), respectively.

We used a linear mixed-effects model (LME) to examine which aspects of the physical environment drive the selection of airspeed, with climb rate, flight altitude and land cover as predictors. The interaction between climb rate and altitude was also included in the global model, to account for the influence of altitude on flight forces. A model comparison showed that the interaction between climb rate and land cover did not improve the model (AIC difference = 1, $\chi^2 = 3.333$, $p = 0.068$), so this interaction was removed from the model. Day of the flight and bird identity were included as random factors in this and subsequent LMEs. Statistical analyses were conducted in R-Studio, using R v. 3.3.2 [43] using the packages 'lme4' v. 1.1-19 [44], 'car' v. 3.0-3 [45] and 'MuMIn' v. 1.43.6 [46]. A visual representation of the GPS tracks was generated using the R package 'ggmap' [47].

## 3. Results

Overall, 88 homing flights were recorded from eight male pigeons (mean mass ± s.d. = 455.0 ± 14.7 g). Once the flights with interruptions, missing data or average wind speeds greater than 2 m s$^{-1}$ were excluded, 29 flights were available for further analysis (15 from 2018 and 14 from 2019; one pigeon was tested in 2019 only). The travelling section of the homing flight lasted an average of 6.1 ± 1.0 min (mean ± s.d.) and covered 7.2 ± 0.9 km (mean ± s.d.). No differences in individual wing loading were observed between the two field seasons (paired $t$-test: $t = 1.456$, $p = 0.219$) and neither was there a significant difference in the average airspeed recorded for each pigeon (paired $t$-test: $t = 0.357$, $p = 0.739$).

Birds flew with a mean airspeed of 19.9 m s$^{-1}$ (± 2.6 s.d.), with speed varying by 10.4 m s$^{-1}$ on average within each flight, and an overall maximum of 23.0 m s$^{-1}$ across individuals. The mean speed was, therefore, substantially higher than the theoretical maximum range speed (mean $V_{mr} = 16.4$ m s$^{-1}$). Nonetheless, $V_{mp}$ (mean = 12.4 m s$^{-1}$) was a good predictor of minimum speeds, as birds rarely flew below $V_{mp}$, even during ascending flight, when speeds were lowest.

Climb rate was the strongest predictor of airspeed ($V_a$), with speed decreasing with increasing climb rate, $V_z$ (table 1 and figure 2). When airspeed was considered separately for climbing and descending

**Table 1.** Statistical results of the LME model showing the effect on airspeed ($V_a$) of the rate of change of altitude ($V_z$), land cover, altitude ASL and the interaction between $V_z$ and altitude (LME model: $R_m^2 = 0.27$, $R_c^2 = 0.62$). The model was executed with standardized (centred and scaled) variables to compare the magnitude of their effects. A higher estimate shows an effect of greater magnitude ($V_z$). Raw estimates (unstandardized) are given in the left column to allow quantitative interpretation of these effects.

| | estimate (unstd.) | estimate (std.) | s.e. | t-value | $\chi^2$ | p-value |
|---|---|---|---|---|---|---|
| (intercept) | 16.182 | 20.089 | 0.311 | 64.624 | NA | NA |
| $V_z$ | −1.239 | −1.323 | 0.017 | −77.579 | 5915.680 | <0.001 |
| land cover (woodlands) | −0.165 | −0.165 | 0.046 | −3.619 | 13.099 | <0.001 |
| altitude | 0.008 | 0.310 | 0.028 | 10.908 | 116.953 | <0.001 |
| $V_z$: altitude | −0.004 | −0.187 | 0.018 | −10.544 | 111.185 | <0.001 |

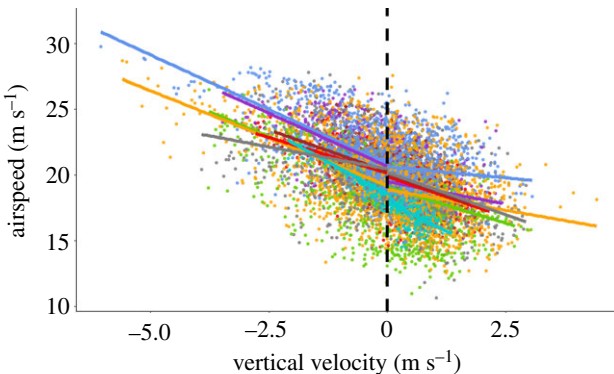

**Figure 2.** Relationship between airspeed and vertical velocity in seven homing pigeons. Each individual is represented in different colours. The dashed line shows the limit between descending (left) and climbing (right).

flight, the relationship between $V_a$ and $V_z$ remained linear, but we observed a better fit and a steeper slope in descending flight compared with ascending flight (LME model, climbing: estimate = −1.093, s.e. = 0.043, $\chi^2 = 657.560$, p-value < 0.001, $R_m^2 = 0.07$, $R_c^2 = 0.53$; descending: estimate = −1.16, s.e. = 0.031, $\chi^2 = 1394.3$, p-value < 0.001, $R_m^2 = 0.14$, $R_c^2 = 0.59$) (figure 2). The effects of land cover, altitude and the interaction between $V_z$ and altitude were also significant, but the difference in airspeed between land cover was minor (0.165 m s$^{-1}$ slower over woodlands, see table 1).

Flight altitude varied between 401 and 630 m ASL (the highest topographical point in the area was 716 m). Birds climbed more rapidly when flying over steeper terrain; however, this explained only 1% of the variation in climb rate (LME model: estimate = 0.06, $\chi^2 = 99.54$, p < 0.001, $R_m^2 = 0.01$, $R_c^2 = 0.01$) (figure 3c,d). A comparison of flight altitude over the plain before the hill showed a significant effect of the subsequent route on the flight altitude; birds that continued along the valley flew on average 51.6 m lower than the birds that flew over the hill (LME model: estimate = −51.58, $\chi^2 = 18.56$, s.e. = 11.98, p < 0.001, $R_m^2 = 0.25$, $R_c^2 = 0.62$).

One of the most striking and unanticipated features was the fine-scale variability in airspeed, as substantial and rapid changes in speed were exhibited during the flights (figure 3a), with accelerations ranging from −4.5 to 3.5 m s$^{-2}$ (median: 0.0, interquartile range (IQR): 0.6 m s$^{-2}$). Altitude was also very variable, with a median climb rate of 0.6 m s$^{-1}$ (IQR: 0.7 m s$^{-1}$), and a median descent rate of − 0.7 m s$^{-1}$ (IQR: 0.9 m s$^{-1}$) when climbing and descending was considered across flights (figure 3b). The maximum climb angle was 14°, with 90% of angles being between 0° and 5°. Variability in climb rate did not differ greatly between valley and hill flights (standard deviation: 1.0 and 1.2 m s$^{-1}$, respectively), and both routes were associated with substantial variation in altitude (figure 3c,d). An ultralight, equipped with the same tagging technology, simultaneously flew a section of the pigeon's flight path with the intention of maintaining a fixed speed and altitude. Data from the ultralight flight

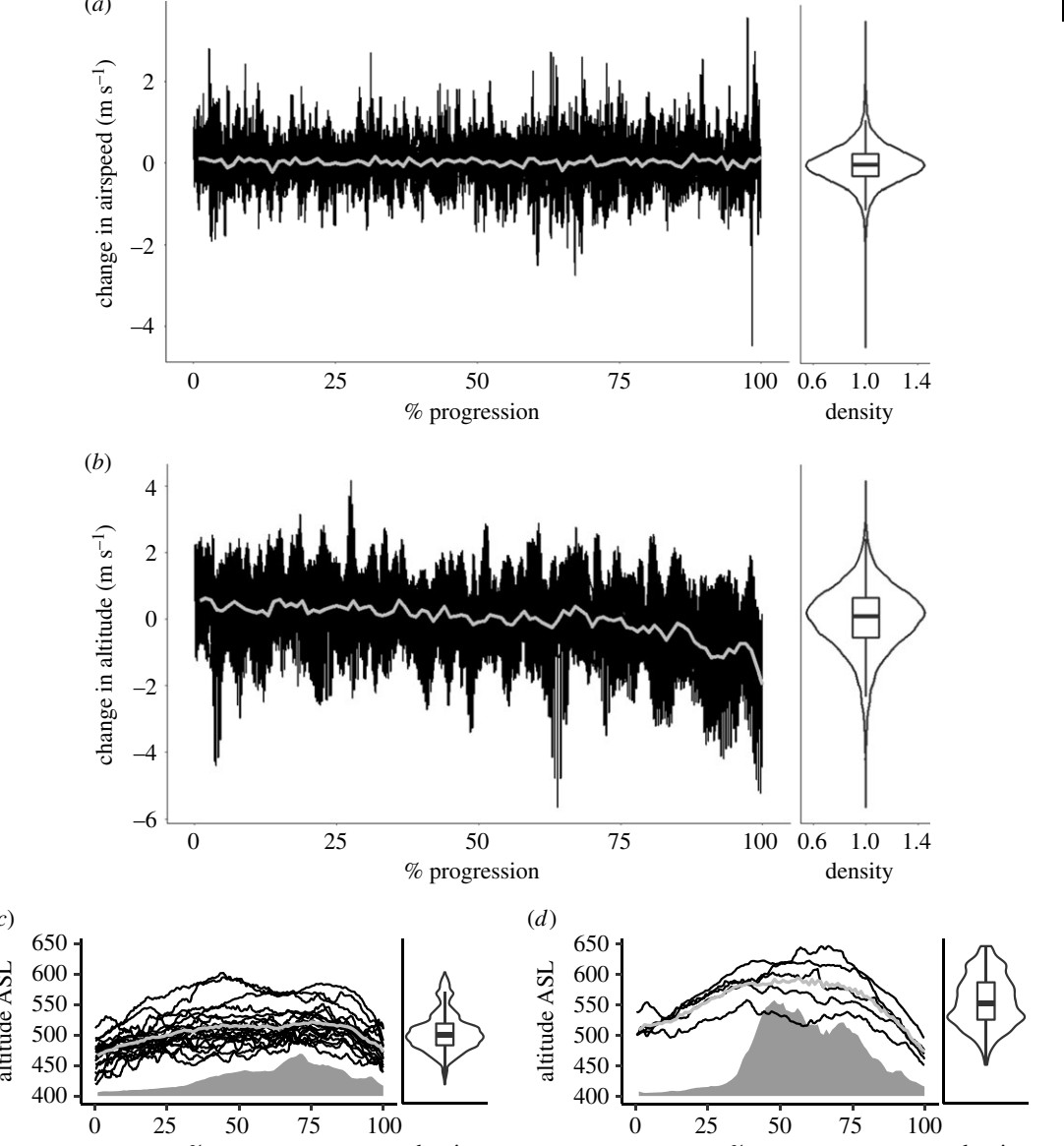

**Figure 3.** Change in (*a*) airspeed per second and (*b*) altitude per second, in relation to the proportion of time through 25 homing flights. (*c*) and (*d*) depict the elevation and altitude profiles of birds flying over the valley (*n* = 20) and above the hill (*n* = 5), respectively. The grey line corresponds to the average value calculated for all tracks. The filled grey area represents the average ground elevation below the birds. Violin plots show the distribution of the data, while box plots show the median, upper and lower quartiles, and the distribution of the data excluding the outliers.

showed markedly less variation in climb rate and speed, suggesting that the variability is a specific feature of pigeon flight (electronic supplementary material).

## 4. Discussion

Homing pigeons have been bred for their ability to return to the loft quickly, and the selective pressure to minimize the flight time is likely to outweigh that to minimize power (in relation to time or distance). In addition, pigeons know that the distance to their loft is short, and may thus be unlikely to employ an energy-efficient flight style. We, therefore, believe that many of our results can be interpreted within this high-performance context, as supported by the high mean flight speeds (some 3.5 m s$^{-1}$ greater than predicted for $V_{mr}$), which are consistent with other studies on homing pigeons [32,48]. Furthermore, flight speeds in this study frequently exceeded $V_{mr}$ even during climbing. These high

speeds contrast with those recorded from wild species with a mass similar to pigeons, which migrate at a speed close to $V_{mr}$ [9].

Our highest speeds occurred in descending flight (as observed, but not quantified, by Tucker [49]), with speeds increasing with steeper descents. This is likely to be due to the conversion of some of the potential energy into aerodynamic power, enabling birds to accrue energy savings for the fastest speeds. The motivation for the extremely high speeds found at the end of some flights is likely to be due to goshawks waiting for pigeons in the vicinity of the loft [50], causing them to descend faster. Nonetheless, the fact that the slope of the relationship between airspeed and vertical velocity varied between climbing and descending flight demonstrates that average speeds taken over entire flights will be biased upwards by periods of descent. In the context of behavioural ecology, this means that estimates of power use, or the currency driving speed selection, cannot be extrapolated from measurements of speed without accounting for changes in altitude.

Despite operating within a high-power framework, pigeons did show some signs of judicious energy use, a prominent example being climbing at minimum rates to fly over high points. Specifically, birds started climbing shortly after the beginning of the flight and approximately 2 km ahead of hills on their route, indicating that they anticipated the topographic change. While gaining height early in the flight may also be advantageous for navigation and reducing predation risk, the distribution of climb rates, which was centred around 0°, with 90% of positive climb angles between 0° and 5° (cf. [8,15]), constitutes a time and/or energy saving. Gradual climbs have also been observed in bar-headed geese [4], which are more limited by energy than our pigeons, suggesting that in general, birds may favour this strategy for energetic reasons.

Nonetheless, the remarkable variation in both altitude and speed observed in flights highlights a major source of energetic inefficiency. Barometric pressure provides the best estimates of relative changes in the altitude at small spatial and temporal scales [41], and both speed and altitude were smoothed over 5 s to remove the variability that could be caused by logger inaccuracy. This strongly suggests that pigeons willingly adopt a variable flight style, a behaviour that was not predicted at the outset. Whether animals aim to optimize their use of time or energy, they should maintain a constant speed and altitude [5] and adopt a path with minimum tortuosity, because turns are energetically costly [48], they increase the overall path length, and accelerations and decelerations will be more costly than simply maintaining a constant speed [51]. In this respect, birds did not present profiles of animals maximizing power for overall homing speed. Specifically, our pigeons exhibited substantial fine-scale variability in speed and rate of change of altitude and took horizontal paths that deviated appreciably from that of a straight line (figure 1), despite training prior to the experimental releases to control for changes in familiarity [52]. Future studies will need to consider changes in route familiarity or experience within and between flights, given that this affects estimates of speed and efficiency at the level of individual flights [32]. While aspects of navigation, such as following landscape features [30], can lead to horizontal track variation, this does not account for the observed variation in height nor for the effect on speed and increased path length that this may have. In our study, the fine-scale changes in altitude amounted to an additional 178.7 m per flight (compared with a vertical profile smoothed over 20 s).

We suggest that the marked, apparently inefficient, variability in pigeon flight patterns may be explained within the context of predator defence. Homing pigeons are common targets for raptors [53], most notably peregrine falcons *F. peregrinus*, sparrowhawks *Accipiter nisus* and goshawks *A. gentilis*, with sources quoting losses during races of up to 23% due to peregrine falcons alone [54]. A study taking place in our study area recorded 15 attacks during 27 flights [50]. It is likely to be relevant that artificial selection by breeders can select for birds to fly faster, but cannot avoid selection pressures related to predators on their routes. Moving in a variable way is a strategy adopted by numerous taxa to avoid, and reduce the accuracy of, predator attacks [55–57]. Such strategies are known as protean behaviours, and they work by preventing predators from predicting the future positions of prey engaging in unpredictable lateral movements and altitude and/ or speed. These movements can occur specifically as a reaction to a defined attack (cf. examples in [55,57]) or occur as constant changes in trajectory even when predators are not immediately apparent [57,58]. Well-known examples include the common snipe (*Gallinago gallinago*) and jack snipe (*Lymnocryptes minimus*) [58]. While birds in this study did not adopt an erratic flight style that was obvious to observers, the variation in both speed and vertical velocity in the high-frequency logger data is notable. Our results suggest that, far from being a distinctive but relatively rare behaviour in birds, the protean movement could be widespread, expressed in the form of fine-scale changes in trajectory.

The archetypal strategy for reducing individual predation risk during flight is flocking [59,60], which also leads to higher flight costs in pigeons [48]. However, flocking is not always possible, for instance, solitary breeders must make solo flights to and from the nest. Individuals, therefore, need a range of strategies to reduce predation risk, including for when they are forced to fly solo or in pairs, when the risk of being caught is higher [59]. While further testing is required to establish the effectiveness of this as an anti-predation strategy, the irregular flight style reported in this study may be a response to predation risk when flying in a large group is not possible. It is unlikely to be tenable or needed in large flocks, where the costs and risks of collision are already high [48]. Further high-frequency data will provide insight into how widespread such behaviours are, and how they vary with the number of flock mates.

In conclusion, pigeons do not seem to primarily adopt energy-efficient strategies that minimize overall cost in returning to their loft. Rather, they use high power to return fast and exhibit ostensibly inefficient behaviour in the form of varying movement in terms of altitude, speed and overall trajectory. We propose that this corresponds to a previously unidentified form of protean behaviour allowing better predator avoidance, with birds offsetting the proximate costs of increased energy expenditure for the ultimate benefit of reducing predation risk. Estimating the cost of variable locomotion is notoriously difficult [51], given that protocols for measuring metabolic costs in controlled conditions are based on steady-state movement. Nonetheless, this may prove an important element in understanding how risk affects flight costs in the wild.

Ethics. The experiments were conducted under animal experiment permit number IP-1718-23 issued by Swansea University AWERB and permit G-17/92 issued by the Regierungspräsidium Freiburg, Baden-Württemberg, Germany.

Data accessibility. Data and code used for the analyses of this manuscript are available from the Dryad Digital Repository (https://doi.org/10.5061/dryad.x69p8czh8) [61]. All data and code are also publicly available on Movebank (https://www.movebank.org/cms/webapp?gwt_fragment=page=studies).

Authors' contributions. The experiment was conceived by E.L.C.S., M.W., M.Q., G.D. and B.G., and conducted by E.L., M.Q. and B.G. Statistical analysis was carried out by B.G. and overseen by L.B. Interpretation of the results was undertaken by B.G., E.L.C.S., R.P.W., S.J.P. and A.H., and B.G., E.L.C.S., R.P.W. and S.J.P. wrote the first draft. All authors contributed to the revision of the manuscript. All authors gave final approval for publication and agree to be held accountable for the work performed therein.

Competing interests. We declare we have no competing interests.

Funding. B.G. and E.L.C.S. are supported by the European Research Council under the European Union's Horizon 2020 research and innovation programme grant no. 715874 (to E.L.C.S.). Fieldwork was also supported by a Max Planck Sabbatical Fellowship to E.L.C.S.

Acknowledgements. We would like to address some special thanks to Mark Holton, Carlo Catoni, Philip Hopkins for the fabrication of the loggers and their housing. We also thank the pigeons' owner, Bernhard Banzer, who kindly gave us access to his pigeons and helped us in the releases, as well as Heidi Schmid for driving us to the release site every morning.

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
