## [Peer Review File · Royal Society Open Science]

Review History

RSOS-210130.R0 (Original submission)

Review form: Reviewer 1

Is the manuscript scientifically sound in its present form?

Yes

Are the interpretations and conclusions justified by the results?

Yes

Is the language acceptable?

Yes

Do you have any ethical concerns with this paper?

No

Have you any concerns about statistical analyses in this paper?

No

Recommendation?

Accept with minor revision (please list in comments)

Comments to the Author(s)

The authors described the flight speed pattern of pigeons released from a very familiar location at 5 km from the loft. They found that their pigeons displayed a high variability in speed, but that the variation in speed was not related to the over-flown landscape features.

The manuscript is interesting and well written. I have only a suggestion concerning the Discussion.

Reading the Discussion one has the impression that the typical behaviour of homing pigeons is described. Actually, this is a special case of overtrained pigeons released from one single very familiar site at short distance from home. The authors should acknowledge that this is a special case, because the same kind of study performed over unfamiliar areas at further distances might reveal a different pattern of flight speed. For instance, a difference in speed might emerge when pigeons from unfamiliar areas encounter familiar areas. Inexperienced homing pigeons might behave differently in comparison to experienced homing pigeons. Importantly, the authors should discuss the possibility that the level of experience of the pigeons and the familiarity with the test area might change the speed pattern in relation to the landscape features.

Review form: Reviewer 2

Is the manuscript scientifically sound in its present form?

No

Are the interpretations and conclusions justified by the results?

No

Is the language acceptable?

Yes

Do you have any ethical concerns with this paper?

No

Have you any concerns about statistical analyses in this paper?

No

Recommendation?

Major revision is needed (please make suggestions in comments)

Comments to the Author(s)

This is an information-rich account detailing the flights of pigeons from a release point to a loft. I enjoyed the MS a lot.

Two high tech loggers were combined to provide 3D positional information at high frequency (1 - 5 Hz). The obvious question is the degree to which measurement error contributes to the moment-to-moment changes shown in Figure 3. Next to no information is provided about the

precision of either the Daily Diary (which measured air pressure) or the GPS unit (position). Would the devices record a similar amount of variability if flown at the same speed and similar route on a drone (which presumably would not show protean movements)? Perhaps other controls were carried out?

This is important because much is made of the variability as an anti-predation tactic. The hypothesis seems logically sound and plausible, but even if small, the measurement error in pressure and position would contribute to moment-to-moment variability, and should not be interpreted as protean behavior.

A second and similar criticism is the question of whether the size and weight of the package (see my comment on line 111 below) could contribute to the magnitude of the protean movements. It's less obvious to me how this could be controlled for, but perhaps the authors have given this some thought and could help allay concern by addressing it specifically.

Minor comments

Lines 18 -19

The flight speeds that animals should adopt to minimise energy expenditure in different scenarios can be predicted by the curve of power against speed.

The power curve is used to predict several optima, not only minimum energy expenditure. A more general opening sentence might be

The power curve provides a basis for predicting adjustments that animals make in flight speed, for example in relation to wind, distance, habitat foraging quality, and objective. However, relatively few studies

Line 111 producing a unit measuring $47 \times 22 \times 15$ cm

Surely this is an error. I am an average-sized white male (sorry about that!) but would sense some awkwardness in maneuvering with a package of this size strapped to my back, even if it weighs only 18g. I assume this must be mm – even so, that's pretty big for a pigeon.

Line 200 - 201

It is not reported how acceleration was calculated. I assume from the context that this is the change in speed between successive 1 second segments of the flight path. So, if the speed over segment 1 is 20 ms⁻¹, and the speed over segment 2 is 18 ms⁻¹, the 'acceleration' is -2 ms⁻²? But perhaps acceleration is one of the parameters measured by the Daily Diary (see lines 106 -107)?

Decision letter (RSOS-210130.R0)

Dear Mr Garde

The Editors assigned to your paper RSOS-210130 "Fine-scale changes in speed and altitude suggest protean movements in homing pigeon flights" have now received comments from reviewers and would like you to revise the paper in accordance with the reviewer comments and any comments from the Editors. Please note this decision does not guarantee eventual acceptance.

Please submit your revised manuscript and required files (see below) no later than 21 days from today's (ie 04-Mar-2021) date. Note: the ScholarOne system will 'lock' if submission of the revision is attempted 21 or more days after the deadline. If you do not think you will be able to meet this deadline please contact the editorial office immediately.

on behalf of Dr Agustina Gómez-Laich (Associate Editor) and Kevin Padian (Subject Editor)
openscience@royalsociety.org

Associate Editor Comments to Author (Dr Agustina Gómez-Laich):
Comments to the Author:
Dear Garde and co-authors,

The manuscript "Fine-scale changes in speed and altitude suggest protean movements in homing pigeon flights" has now been seen by two reviewers, both of whom found the work novel, interesting and well written. However, both raised some concerns and present suggestions as to how this contribution could be improved.

Reviewer#1 suggests to acknowledge that the observed flight speed pattern corresponds to a special case of study and may not be the typical behaviour of homing pigeons. Additionally, this reviewer suggests authors to discuss the possibility that the level of experience of the pigeons and the familiarity with the area might change the speed pattern in relation to the landscape features. Reviewer#2 points out whether and how logger precision might have contributed to the moment-to-moment variability. Even though the anti-predation tactic hypothesis seems logically sound and plausible, the measurement error in pressure and position would contribute to the moment-to-moment variability. Finally, this reviewer also raises the question of whether the size and weight of the deployed devices could contribute to the magnitude of the protean movements.

Reviewer comments to Author:

Reviewer: 1

Comments to the Author(s)

The authors described the flight speed pattern of pigeons released from a very familiar location at 5 km from the loft. They found that their pigeons displayed a high variability in speed, but that the variation in speed was not related to the over-flown landscape features.

The manuscript is interesting and well written. I have only a suggestion concerning the Discussion.

Reading the Discussion one has the impression that the typical behaviour of homing pigeons is described. Actually, this is a special case of overtrained pigeons released from one single very familiar site at short distance from home. The authors should acknowledge that this is a special case, because the same kind of study performed over unfamiliar areas at further distances might reveal a different pattern of flight speed. For instance, a difference in speed might emerge when pigeons from unfamiliar areas encounter familiar areas. Inexperienced homing pigeons might behave differently in comparison to experienced homing pigeons. Importantly, the authors should discuss the possibility that the level of experience of the pigeons and the familiarity with the test area might change the speed pattern in relation to the landscape features.

Reviewer: 2

Comments to the Author(s)

This is an information-rich account detailing the flights of pigeons from a release point to a loft. I enjoyed the MS a lot.

Two high tech loggers were combined to provide 3D positional information at high frequency (1 - 5 Hz). The obvious question is the degree to which measurement error contributes to the moment-to-moment changes shown in Figure 3. Next to no information is provided about the precision of either the Daily Diary (which measured air pressure) or the GPS unit (position). Would the devices record a similar amount of variability if flown at the same speed and similar route on a drone (which presumably would not show protean movements)? Perhaps other controls were carried out?

This is important because much is made of the variability as an anti-predation tactic. The hypothesis seems logically sound and plausible, but even if small, the measurement error in pressure and position would contribute to moment-to-moment variability, and should not be interpreted as protean behavior.

A second and similar criticism is the question of whether the size and weight of the package (see my comment on line 111 below) could contribute to the magnitude of the protean movements. It's less obvious to me how this could be controlled for, but perhaps the authors have given this some thought and could help allay concern by addressing it specifically.

Minor comments

Lines 18 -19

The flight speeds that animals should adopt to minimise energy expenditure in different scenarios can be predicted by the curve of power against speed.

The power curve is used to predict several optima, not only minimum energy expenditure. A more general opening sentence might be

The power curve provides a basis for predicting adjustments that animals make in flight speed, for example in relation to wind, distance, habitat foraging quality, and objective. However, relatively few studies

Line 111 producing a unit measuring $47 \times 22 \times 15$ cm

Surely this is an error. I am an average-sized white male (sorry about that!) but would sense some awkwardness in maneuvering with a package of this size strapped to my back, even if it weighs only 18g. I assume this must be mm – even so, that's pretty big for a pigeon.

Line 200 - 201

It is not reported how acceleration was calculated. I assume from the context that this is the change in speed between successive 1 second segments of the flight path. So, if the speed over segment 1 is 20 ms⁻¹, and the speed over segment 2 is 18 ms⁻¹, the 'acceleration' is -2 ms⁻²? But perhaps acceleration is one of the parameters measured by the Daily Diary (see lines 106 -107)?

===PREPARING YOUR MANUSCRIPT===

===PREPARING YOUR REVISION IN SCHOLARONE===

Author's Response to Decision Letter for (RSOS-210130.R0)

See Appendix A.

Decision letter (RSOS-210130.R1)

Dear Mr Garde

On behalf of the Editors, we are pleased to inform you that your Manuscript RSOS-210130.R1 "Fine-scale changes in speed and altitude suggest protean movements in homing pigeon flights" has been accepted for publication in Royal Society Open Science subject to minor revision in accordance with the referees' reports. Please find the referees' comments along with any feedback from the Editors below my signature.

Please submit your revised manuscript and required files (see below) no later than 7 days from today's (ie 31-Mar-2021) date. Note: the ScholarOne system will 'lock' if submission of the revision is attempted 7 or more days after the deadline. If you do not think you will be able to meet this deadline please contact the editorial office immediately.

on behalf of Dr Agustina Gómez-Laich (Associate Editor) and Kevin Padian (Subject Editor)
openscience@royalsociety.org

Associate Editor Comments to Author (Dr Agustina Gómez-Laich):

Dear authors,

I have only a few minor comments and suggestions that are listed below. Specific comments relate to the page and line number of the clean version of the word document that was available for review.

Methods.

Line 109. Please incorporate a space after "±"

Legend Figure 1.

I suggest mentioning which R package was used in the methodology instead of in the figure legend.

Supplementary Information

I suggest incorporating the data from the tag placed on the ultralight.

Reviewer comments to Author:

===PREPARING YOUR MANUSCRIPT===

===PREPARING YOUR REVISION IN SCHOLARONE===

Author's Response to Decision Letter for (RSOS-210130.R1)

See Appendix B.

Decision letter (RSOS-210130.R2)

Dear Mr Garde,

I am pleased to inform you that your manuscript entitled "Fine-scale changes in speed and altitude suggest protean movements in homing pigeon flights" is now accepted for publication in Royal Society Open Science.

on behalf of Dr Agustina Gómez-Laich (Associate Editor) and Kevin Padian (Subject Editor)
openscience@royalsociety.org

Appendix A

Editor's comments

The manuscript "Fine-scale changes in speed and altitude suggest protean movements in homing pigeon flights" has now been seen by two reviewers, both of whom found the work novel, interesting and well written. However, both raised some concerns and present suggestions as to how this contribution could be improved.

We are delighted the reviewers found our study interesting and well written. Please find our response to your comments below in blue.

Reviewer: 1

The authors described the flight speed pattern of pigeons released from a very familiar location at 5 km from the loft. They found that their pigeons displayed a high variability in speed, but that the variation in speed was not related to the over-flown landscape features.

The manuscript is interesting and well written. I have only a suggestion concerning the Discussion.

Thank you for the positive feedback on our study.

Reading the Discussion one has the impression that the typical behaviour of homing pigeons is described. Actually, this is a special case of overtrained pigeons released from one single very familiar site at short distance from home. The authors should acknowledge that this is a special case, because the same kind of study performed over unfamiliar areas at further distances might reveal a different pattern of flight speed. For instance, a difference in speed might emerge when pigeons from unfamiliar areas encounter familiar areas. Inexperienced homing pigeons might behave differently in comparison to experienced homing pigeons. Importantly, the authors should discuss the possibility that the level of experience of the pigeons and the familiarity with the test area might change the speed pattern in relation to the landscape features.

We are a bit confused about what is meant by "over-trained". Homing studies tend to focus either on the changes that occur when the route is being established, or remove the influence of learning by assessing behaviour once the route has been fixed. We took the latter approach and have added detail to the methods on this point, as well as additional references where a similar number of flights have been used (lines 101-107 of the version with tracked changes).

We agree that different results might emerge if we were to vary the route familiarity, which is something that would require a follow on study. To the best of our knowledge, the only study to address the issue of how route familiarity affects flight performance, in terms of parameters beyond overall tortuosity and speed, is that by Taylor et al. (2017). As the reviewer notes, the findings of this study could be extrapolated to predict changes in speed along the route for scenarios where familiarity varies, and we have added text in the discussion to make this clear (lines 275-277).

Finally, the fine scale of the variance that we report in our study, and the fact that this occurs continuously in flights where we control for familiarity, suggests that this is not related to speed changes.

Reviewer: 2

This is an information-rich account detailing the flights of pigeons from a release point to a loft. I enjoyed the MS a lot.

Many thanks for your encouraging feedback.

Two high tech loggers were combined to provide 3D positional information at high frequency (1 - 5 Hz). The obvious question is the degree to which measurement error contributes to the moment-to-moment changes shown in Figure 3. Next to no information is provided about the precision of either the Daily Diary (which measured air pressure) or the GPS unit (position). Would the devices record a similar amount of variability if flown at the same speed and similar route on a drone (which presumably would not show protean movements)? Perhaps other controls were carried out?

This is important because much is made of the variability as an anti-predation tactic. The hypothesis seems logically sound and plausible, but even if small, the measurement error in pressure and position would contribute to moment-to-moment variability, and should not be interpreted as protean behavior.

Thank you for your comment, device-related error is a pertinent point and something that has been considered in detail in a publication that includes some of our authors (Péron et al. 2020). In the manuscript we now add details of the specific pressure sensor and its accuracy (± 1 m) (lines 114-115). We feel confident that this, combined with the smoothing of the raw pressure values over 5 s (see line 138) to minimize the effect of any residual sensor noise, means that we are reporting a biologically meaningful result.

It's interesting that the reviewer mentions the idea of a drone. We do have data from a tag placed on an ultralight that flew the same flight route as the pigeons while trying to maintain a constant altitude and speed. The resulting pressure data show much more consistency than the data from the pigeons. Given the relative accuracy reported for the barometric pressure sensor and the smoothing of the data we don't feel that adding data from the ultralight would add to the manuscript, as the outcomes in terms of data processing would remain the same. Nonetheless, we would be happy to include this in the supplementary information if the editor feels it would be useful.

GPS inaccuracy could contribute to errors in the calculation of flight speed, but should be minimal, as a resolution of 1 Hz should provide an accuracy of 2 m for GiPSy 5 loggers in open areas. Nonetheless, we also smoothed speed over 5 s to reduce the error. We now address this in the discussion (lines 264-268).

A second and similar criticism is the question of whether the size and weight of the package (see my comment on line 111 below) could contribute to the magnitude of the protean movements. It's less obvious to me how this could be controlled for, but perhaps the authors have given this some thought and could help allay concern by addressing it specifically.

It's true that the tag mass and dimensions are on the high side when considered in relation to what you would attach to free-living animals. Nonetheless, we feel that they are within acceptable limits, being between 3.8 and 4.2 % of a bird's body mass (less than devices used in other recent studies that collected similarly high resolution data on homing pigeons e.g. Sankey et al 2019, Taylor et al 2017).

It is an interesting question of whether the variability in flight parameters could represent a tagging effect. However, while loggers would be predicted to increase flight effort through the associated increase in mass and drag, we agree that there is no clear mechanistic link between tag mass and an increase in fine-scale variability in speed and altitude (in fact if anything you might predict the opposite to what we report, with tagged animals being less likely to engage in such costly behaviour).

Lines 18 -19

The flight speeds that animals should adopt to minimise energy expenditure in different scenarios can be predicted by the curve of power against speed. The power curve is used to predict several optima, not only minimum energy expenditure. A more general opening sentence might be “The power curve provides a basis for predicting adjustments that animals make in flight speed, for example in relation to wind, distance, habitat foraging quality, and objective. However, relatively few studies

Thanks for the suggestion. We have changed the opening sentence accordingly (lines 18-21).

Line 111 producing a unit measuring $47 \times 22 \times 15$ cm

Surely this is an error. I am an average-sized white male (sorry about that!) but would sense some awkwardness in maneuvering with a package of this size strapped to my back, even if it weighs only 18g. I assume this must be mm – even so, that’s pretty big for a pigeon.

Thanks for spotting this! The size was in mm indeed, and this has now been corrected (line 119).

Line 200 - 201

It is not reported how acceleration was calculated. I assume from the context that this is the change in speed between successive 1 second segments of the flight path. So, if the speed over segment 1 is 20 ms^{-1} , and the speed over segment 2 is 18 ms^{-1} , the ‘acceleration’ is -2 ms^{-2} ? But perhaps acceleration is one of the parameters measured by the Daily Diary (see lines 106 -107)?

Apologies for the omission. We have now added information to explain that acceleration is calculated from speed (line 153-155).

Appendix B

Authors' point-by-point response to Editors' comments

Article name: Fine-scale changes in speed and altitude suggest protean movements in homing pigeon flights

Authors: Baptiste Garde, Rory P. Wilson, Emmanouil Lempidakis, Luca Börger, Steven J. Portugal, Anders Hedenström, Giacomo Dell'Omo, Michael Quetting, Martin Wikelski, Emily L. C. Shepard.

I have only a few minor comments and suggestions that are listed below. Specific comments relate to the page and line number of the clean version of the word document that was available for review.

Thank you for the positive feedback on our study. Please find our response to your comments below in blue.

Methods.

Line 109. Please incorporate a space after “±”

This was corrected.

Legend Figure 1.

I suggest mentioning which R package was used in the methodology instead of in the figure legend.

The information about the package used was moved to Methods (lines 160-161)

Supplementary Information

I suggest incorporating the data from the tag placed on the ultralight.

We have incorporated a new figure in the Supplementary information showing the variation in altitude and speed in simultaneous flights recorded from an ultralight and a pigeon. We have added a sentence referring to this figure in the main text (lines 216-220).